# BERT for Activity Recognition Using Sequences of Skeleton Features and Data Augmentation with GAN

**DOI:** 10.3390/s23031400

**Published:** 2023-01-26

**Authors:** Heilym Ramirez, Sergio A. Velastin, Sara Cuellar, Ernesto Fabregas, Gonzalo Farias

**Affiliations:** 1Escuela de Ingeniería Eléctrica, Pontificia Universidad Católica de Valparaíso, Av. Brasil 2147, Valparaíso 2362804, Chile; 2School of Electronic Engineering and Computer Science, Queen Mary University of London, London E1 4NS, UK; 3Department of Computer Science and Engineering, Universidad Carlos III de Madrid, 28912 Madrid, Spain; 4Departamento de Informática y Automática, Universidad Nacional de Educación a Distancia, Juan del Rosal 16, 28040 Madrid, Spain

**Keywords:** activity recognition, human skeleton, pose estimation, BERT, computer vision

## Abstract

Recently, the scientific community has placed great emphasis on the recognition of human activity, especially in the area of health and care for the elderly. There are already practical applications of activity recognition and unusual conditions that use body sensors such as wrist-worn devices or neck pendants. These relatively simple devices may be prone to errors, might be uncomfortable to wear, might be forgotten or not worn, and are unable to detect more subtle conditions such as incorrect postures. Therefore, other proposed methods are based on the use of images and videos to carry out human activity recognition, even in open spaces and with multiple people. However, the resulting increase in the size and complexity involved when using image data requires the use of the most recent advanced machine learning and deep learning techniques. This paper presents an approach based on deep learning with attention to the recognition of activities from multiple frames. Feature extraction is performed by estimating the pose of the human skeleton, and classification is performed using a neural network based on Bidirectional Encoder Representation of Transformers (BERT). This algorithm was trained with the UP-Fall public dataset, generating more balanced artificial data with a Generative Adversarial Neural network (GAN), and evaluated with real data, outperforming the results of other activity recognition methods using the same dataset.

## 1. Introduction

In recent years, the detection and recognition of human activities through machine learning algorithms have become a field of significant interest to the scientific community. As most human beings live in highly social environments, the interest arises because such research can be used in applications such as personal security and safety, healthcare, assistance to the elderly, sports, and so on.

Many of the approaches for the recognition of human activities presented in the literature have proposed the use of body sensors such as gyroscopes, accelerometers, barometers, life sign measuring devices, etc. These types of sensors are often integrated into cell phones, necklaces, or smartwatches that can detect sudden changes in a person’s movement when they fall or perform a certain movement [1]. The main disadvantage of such approaches is that subjects must wear or place these devices on them. As such, they can be uncomfortable and clearly do not scale up for their use in open spaces and for multiple subjects [2]. More recent proposals that use video images from RGB cameras to detect human activity free subjects from having to carry on-body devices and can be extended to the analysis of human activity in public areas and involving interactions among different people [3,4,5,6].

In other recent cases, instead of using RGB images directly, they are used to extract poses represented by a set of body joints and their interconnection (i.e., human skeleton representation) [5,7,8,9,10], which are then used as features for further analysis. Such skeleton representations have been found powerful to differentiate between different types of activities such as walking, sitting, jumping, running, falling down, etc. Modern posture extraction from RGB images is not limited to images with single individuals, which is an important advantage compared to other works [4]. Indeed, we have used this approach in a previous paper [11], which uses skeleton features with well-known machine learning (ML) methods, demonstrating state-of-the-art performance for public reference datasets. That work also presented tests with an LSTM (Long Short-Term Memory) recurrent neural network that resulted in low performance due to the imbalanced dataset.

The use of deep learning applied to Human Activity Recognition has recently grown. The most popular architectures are Convolutional Neural Networks (CNNs) for their capacity to learn special features from images [12] and Recurrent Neural Networks (RNNs) which are able to learn long-term temporal patterns present in the data [13]. Yadav et al. [14] present an activity recognition and fall detection network (ARFDNet) where the videos are passed to a pose estimation network to extract skeleton features, which are processed and inputted to a CNN followed by gated recurrent units (GRUs) to learn spatiotemporal dynamics, obtaining an accuracy of 96.7%. Song et al. [15] propose a model using different levels of attention with an LSTM to learn discriminative skeleton joints. The work presented in [16] develops a hybrid model by incorporating CNN and LSTM for activity recognition and tests it on a generated dataset, obtaining an accuracy of 90.89%, which shows that the proposed model is suitable for human activity recognition applications. In [17], the visual attention mechanism is used and an end-to-end, two-stream, attention-based LSTM architecture is proposed for action recognition in videos, which can learn detailed spatiotemporal attention features and also can explicitly allocate content and temporal-dependent attention to the output of each deep feature in the video sequence. This method obtained an accuracy of 94.1% on UFC11, 96% on UFC sports, and 69.8% on jHMDB datasets. Other works such as [18] propose a framework that employs deep learning and swarm-intelligence-based optimization techniques with 3D skeleton data for action classification; they extract features such as distance, distance velocity, angle, and angle velocity on encoded images that were fed into a CNN model, which is a modified version of Inception-ResNet, and were evaluated on public datasets achieving an accuracy of 98.13% on UTD-MHAD, 90.67% on HDM05, and 85.45% on NTU RGB+D60. Finally, [19] introduces a soft attention mechanism in Temporal Segment Networks (TSN), which improves the ability to learn long-term information, enables the network to adaptively focus on key features in space and time, and verify the effectiveness on their model in four public datasets achieving an accuracy of 93.3% on UCF101, 67.9% on HMDB51, and 78.4% on JHMDB.

A characteristic of public datasets such as UP-Fall [20] and UR-Fall is that both were created by simulating controlled activities in laboratory conditions and the human activity classes tend to be unbalanced, especially for activities such as falling down. It is well-known that the performance of deep classification algorithms can deteriorate when the data are not balanced—that is, when the available data are not evenly distributed among the different classes. The traditional and best-known approach to mitigate this problem is to increase the dataset by introducing instances corresponding to the classes with minority data, applying geometric transformations to the original instances. For example, in images for this, such transformation translates into rotations or reflections of the pixels. The main disadvantage of this augmentation approach is that it can corrupt relevant orientation-related features. In this sense, the inclusion of synthetic data generation algorithms based on deep learning has had greater importance as an approach to reducing the lack of balance. Generative Adversarial Neural Networks (GANs) have been proposed as a tool to artificially generate realistic data [21,22,23].

The main contributions of the proposed work are as follows:The use of novel deep neural networks such as BERT transformer networks for the recognition of activities using data of skeleton poses extracted from video images. As far as we know, our approach is the first activity recognition model that uses BERT networks directly with numerical data instead of text. Our proposal suggests that it is possible to use the characteristics of the pose (skeleton) of a person in video images as if it were a sentence (text)—this is how BERT interprets it. The way in which we approach this challenge is described in Section 3.2.The development of a direct comparison between the use of machine learning and deep learning models for the recognition of activities using the same methodology and the same dataset.The use of data augmentation of the activities corresponding to the unbalanced classes demonstrating the hypothesis that it is possible to increase the performance of the BERT model by increasing the data of the unbalanced classes of the UP-Fall set using generative networks. The generation of these artificial data is achieved using GANs, which lead to better activity recognition performance. Performance is then compared with our previously published results for the UP-Fall dataset.

The paper is organized as follows. Section 2 presents the description of the UP-Fall database. Section 3 introduces the activity recognition approach and describes AlphaPose, the feature extraction method, the model to use, and the data augmentation system. Section 4 shows the experimental results and the comparison with previous results. Finally, Section 5 presents the conclusions and the proposed future work.

## 2. Datasets

The present work uses the UP-Fall dataset for the recognition of activities. The experimental results are compared with the results obtained in [11] using the same dataset.

As described in Martínez-Villaseñor et al. [20]—for completeness, outlined also here—UP-Fall includes 12 human activities of daily living performed by 17 subjects in a controlled environment, each subject making three repetitions of each activity. The first five activities correspond to fall activities, such as falling forward using hands, falling forward using knees, falling backward, falling sideways, and falling sitting on a flat chair. The other seven activities include more normal activities such as walking, standing, sitting, picking up an object, jumping, lying down, and kneeling. The twelve activities are shown in Table 1. UP-Fall has a total of 816 videos, of which 255 correspond to falls and 561 to daily activities. The images are located at (https://sites.google.com/up.edu.mx/har-up/ accessed on 20 January 2023). The files are organized into 17 folders, one for each subject. Within each folder, there are 11 sub-folders, one for each activity. Within these sub-folders, there are three other sub-folders, one for each repetition. In each sub-folder, there is a CSV file that points to a ZIP file with the recorded images. The complete dataset has 220,660 samples (100%), where 49,544 samples (22.45%) are labeled with fall activities and 171,116 samples (77.55%) are labeled with daily activities.

## 3. Methodology

This work focuses on the recognition of human activities using only video data since, in practical applications such as assisted living and public space monitoring, the use of wearable devices and other sensor modalities is not realistic or convenient. The main hypothesis is that the performance of the methodology provided in our previous work [11] can be verified using articulated bodies (skeletons) extracted from the video, even when modern deep learning techniques are used. Therefore, the goal is to implement an activity recognition method using the UP-Fall dataset and the sliding window methodology described in [11]. The results are obtained when using a Transformers BERT network. After this, GAN networks are tested for data augmentation and the BERT model is re-evaluated with more balanced data.

The method developed for this study is illustrated in Figure 1. It consists of data collection; feature extraction using human skeleton estimation; skeleton filtering; creation of sliding windows (each window with a size of 2 seconds and 3 frames per window); and finally, a neural network classification for recognition activities. All steps are implemented using Python 3.6.

### 3.1. Feature Selection and Extraction

As mentioned in Section 2, the UP-Fall dataset is used to carry out the experiments of this work. This dataset is selected to make a more direct comparison with the work carried out in Ramirez et al. [11]. Once all the UP-Fall images are downloaded, the skeletonization is performed as explained below.

#### 3.1.1. Skeleton Detection

AlphaPose is an open-access method for estimating the postures of multiple people [24], available at (https://www.mvig.org/research/alphapose.html accessed on 20 January 2023). It uses RGB images as input and performs posture detection with a pre-trained model (with the COCO database), obtaining as output the positions (x,y) of 17 key points or joints with coordinates (x,y), together with the detection confidence score of each one, forming a skeleton with the posture of the person or persons of interest. For each person, there are 51(17∗(2+1)) attributes per frame. Time sequences containing these skeleton attributes are generated in this way for UP-Fall.

#### 3.1.2. Sliding Windows

For a direct comparison with [11], sliding windows with the same characteristics and the same dataset are used. Each window has a size of 2 seconds, with 36 frames for each window, of which only 3 frames (the first, the middle, and the last) are selected to reduce the size of each vector of features and to decrease processing time. Therefore, each feature vector with length 153 (51 skeleton attributes * 3 frames) serves as input to the classification model.

Activity recognition is performed using multi-class classifiers. Each sliding window contains one of twelve UP-Fall activities. Of the 207,497 sliding windows, 1473 (0.71%) correspond to the activity of falling forward using hands; 1473 (0.71%) to falling forward using knees; 1858 (0.90%) to falling backward; 1560 (0.75%) to falling sideways; 1863 (0.90%) to falling sitting in an empty chair; 38,570 (18.59%) to walking; 51,573 (24.85%) to standing up; 45,439 (21.90%) to sitting; 1456 (0.70%) to picking up an object; 22,067 (10.63%) to jumping; 38,771 (18.69%) to laying; and 1394 (0.67%) to unknown activity.

### 3.2. Classification Model

This work seeks to improve the results of the best Machine Learning model (RF, Random Forest) reported in [11] with the CAM (camera) modality and the sliding windows methodology using skeletons. Therefore, a deep learning model is chosen to investigate whether that methodology also works with more modern techniques.

Considering the poor results reported in [11] when using an LSTM network, it is decided to test a transformers network with the hypothesis that activity recognition is possible when using recurrent neural networks using the same methodology and the same dataset.

Taking into account that the feature can be represented by one dimension, a Bidirectional Encoder Representation of Transformers (BERT) is used to classify the activities. BERT is a cutting-edge, attention-based natural language processing (NLP) technique [25] created and published by Google in 2018. Bidirectional means that it looks at the left and right context to understand a text. It can be used for prediction, answering questions, language inference, and more. Here, BERT is used for activity classification using the Hugging Face library of transformers. All tests are implemented using Pytorch.

BERT is basically an Encoder stack of transformer architecture. A transformer architecture is an encoder–decoder network that uses self-attention on the encoder side and attention on the decoder side. BERT has 12 layers in the Encoder stack. The BERT architecture has large feedforward networks (768 hidden units). It contains 512 hidden units and 8 attention heads. BERT contains 110 M parameters.

BERT can take sentences as input. The [CLS] token always appears at the start of the text and is specific to classification tasks. The [SEP] token always appears at the end. It is also important to note that the maximum size of tokens that can be fed into the BERT model is 512. If the tokens in a sequence are less than 512, padding is used to fill the unused token slots with [PAD] tokens.

The BERT then passes the input to the above layers. Each layer applies self-attention, passes the result through a feedforward network, and then hands off to the next encoder. For the text classification task, we focus our attention on the embedding vector output from the special [CLS] token. This means that we are going to use the embedding vector of size 768 from the [CLS] token as an input for the classifier, which will then output a vector of the size of the number of classes in the classification task.

In order to convert the words into numerical representations, we first take the sentence and tokenize it. After that, we convert the sentence from a list of strings into a list of numerical indices (word embedding). Thus, the original word is split into smaller subwords and characters. This is because the BERT vocabulary is fixed to a size of 30,000 tokens.

Finally, we need to convert our data to tensors (the input format for the model) and call the BERT model. Thus, this trained vector can be used to perform different kind of tasks such as classification.

Having outlined how BERT works and what it expects at the input, we give way to the novelty of our proposal. Taking into account that our feature vector, which has a size of 153 skeleton features for each sliding window, is made up of numerical data and not of words, our proposal is to eliminate the stage in which BERT tokenizes the sentence (text) and immediately skip to the embedding stage (see Figure 2).

BERT is trained with 30,000 natural language words; so, each word of the linear vector will be represented by a number between 0 and 30,000. Taking that into account, we process our 153 × 1 size features vector as if it were text; thus, all vector data are normalized so that each numeric datum can only have an integer value between 0 and 30,000. Considering that the features of the skeleton’s pose are presented as numerical data (*x* and *y* coordinates in the image plane and a number indicating the score), each feature value maps one of the 30,000 BERT Tokens. To this end, we have used the Min–Max scaling algorithm, which is summarized by the following equation:(1)$$F_{tokenized}=30,000*\frac{F-F_{min}}{F_{max}-F_{min}}$$
where *F* is a feature’s value, and $F_{min}$ and $F_{max}$ are the minimum and maximum numeric values of the entire dataset, respectively. Note that we only consider the integer part of $F_{tokenized}$.

For this work, the hidden_states has four dimensions in the following order:The layer number (13 layers): 13 because the first element is the input embedding; the rest are the outputs of each of BERT’s 12 layers.The batch number: 1 vector (for each sliding window).The word/token number (153 tokens in our vector).The hidden unit/feature number (768 features, BERT’s default).

Table 2 summarizes the hyperparameter settings for the classifier model. The proposed approach uses a pre-trained BERT neural network, which is invoked with the BertModel from _pretrained instruction of PyTorch without modifying any default parameters; these parameters are as follows:input_ids: typing.Optional[torch.Tensor] = None.attention_mask: typing.Optional[torch.Tensor] = None.token_type_ids: typing.Optional[torch.Tensor] = None.position_ids: typing.Optional[torch.Tensor] = None.head_mask: typing.Optional[torch.Tensor] = None.inputs_embeds: typing.Optional[torch.Tensor] = None.labels: typing.Optional[torch.Tensor] = None.next_sentence_label: typing.Optional[torch.Tensor ] = None.output_attentions: typing.Optional[bool] = None.output_hidden_states: typing.Optional[bool] = None.return_dict: typing.Optional[bool] = None.
In addition, it is set with the following options:self.drop = nn.Dropout(p = 0.1),self.out = nn.Linear(self.bert.config.hidden_size, n_classes),
where n_classes = 12.

Our previous work [11] mentioned that an LSTM resulted in low performance due to the notable data imbalance between the 12 UP-Fall classes. That is why in this work it is decided to investigate a data augmentation method to try to balance the classes and to compare the performance between a BERT model with unbalanced data versus a BERT model with balanced data.

### 3.3. Synthetic Data Generation

Generative adversarial networks (GANs) are composed of two deep neural networks: the generator and the discriminator. The goal of the generator is to generate false (artificial) instances that cannot be easily distinguished from true instances by the discriminator [26]. These two networks are trained simultaneously with adverse targets. The discriminator tries to maximize its classification accuracy (by correctly identifying which images came from the generator), while the generator tries to get the discriminator wrong. Recent architectures such as StyleGAN are capable of producing images that are not far from reality [27]. However, most datasets used in the industry are tabular in nature. In this particular approach, the data available to train the model consist of instances composed of vectors of 153 features (51∗3 frames); so, a generation system that is compatible with this type of data is required. Tabular generative models allow the production of artificial data with a distribution similar to the training data, taking into account that these data correspond to a table where each row or instance is sampled independently and each column can contain continuous or discrete values. While in the literature there are several methods to perform this task [28,29], in this work, TABGAN [30] will be used since the model is readily available as a library.

TABGAN contains a continuous data normalization phase in which a variational Gaussian mixture model (VGM) is used to create a vector that encodes each of the continuous instances. Further, the generative network includes a conditional vector, which forces the generator to produce an instance of a specific category. The conditional vector contains all the coded discrete columns with value 0, except the one that we want to satisfy with the generated instance. The training is performed by sampling. In each iteration, a column is randomly selected; from this column, a category is selected based on a probability function built from the frequency of each category in that discrete column. Finally, this category is transformed into the conditional vector that is the input of the generator. This training mode allows the generated distributions to match the distributions of the discrete variables in the training data.

The 207,497 samples were obtained from the UP-Fall dataset, each containing 153 features, with Table 3 showing the data distribution by class. The second and third columns show the number of actual data per class and the equivalent percentage, respectively. It can be seen that classes 6, 7, 8, 10, and 11 contain more than 94% of the data and that the rest of the classes each contain less than 1% of the total. Consequently, models trained on these data may be limited to mostly recognize the majority classes correctly. The fourth column shows the number of artificially generated data for the minority classes, where the number of data is increased by about 40%. It can be seen in the sixth column that the percentages of each class are more similar than before; therefore, the dataset is now more balanced.

Figure 3 shows how the real data are correlated with the false ones. The mean is on the left, the standard deviation is on the right, and the diagonal lines mean equality between real and false data. Each point represents an instance, and it can be seen that they are grouped at two extremes of the graphs. However, they are close to the diagonal line of equality, so it can be said that the data generated are correct. For each feature, it can be seen how similar the real data are to the generated data. Figure 4 shows the cumulative sum of all real and synthetic or fake instances per feature. For clarity, only the first 8 features of the 153 features are shown. It can be seen that characteristics 2 and 5 are the ones that present the most complexity when generated due to their distribution. The graphs are not identical but there are characteristics such as 1, 3, 4, 6, and 7 whose accumulated sums are quite similar for the case of real and fake data.

## 4. Experimental Results

For direct comparison, this paper uses the same performance metrics used in [11], i.e., accuracy, precision, sensitivity, specificity, and $F_{1}$-Score.

To investigate the performance of the sliding window approach using skeleton features and to check if performance can be improved using a BERT transformer network, the experiments developed in this research are as outlined in Section 3.

### 4.1. BERT

The activity recognition method is evaluated using the multi-classifier model with the sequences of the features extracted from the skeleton coordinates, obtained with AlphaPose, for each sliding window in the dataset. The dataset contains 207,497 sliding windows, of which 8227 correspond to fall activities and 199,270 to daily activities. Experiments were performed using 70% of the data for training, 15% for validation, and the remaining 15% for testing.

Table 4 shows the results obtained using BERT. It is observed that the performance of the model notably exceeds the performance of the LSTM reported in [11] that delivered an Accuracy = 81.14%, Precision = 27.76%, Recall = 31.82%, and $F_{1}$-Score = 29.53%. On the other hand, the transformers BERT network delivers Accuracy = 99.14%, Precision = 81.53%, Recall = 80.60%, and $F_{1}$-Score = 80.95%. Therefore, the hypothesis that activity recognition is possible by means of recurrent neural networks trained with sliding windows of skeleton sequences using UP-Fall is demonstrated. It is interesting to note that the machine learning method (RF) presented by [11] (RF model) still delivers better performance with Accuracy = 99.91%, Precision = 97.73%, Recall = 95.60%, and $F_{1}$-Score = 96.63%. It is possible that, as mentioned in [11], the neural network is affected by class imbalance.

Figure 5 shows the confusion matrix when using BERT. It is possible to observe that the classes that tend to have higher cases of confusion are the activities that have the least number of data (less than 205) versus the activities that cover the largest number of the dataset (more than 3000 data).

### 4.2. BERT+TABGAN

As mentioned earlier, and based on the results using the BERT network for activity recognition, it is decided to use the synthetic data generated with a TABGAN network in Section 3.3 for data augmentation with the aim of balancing the number of data per class in the dataset of skeleton sequences.

Figure 6 shows the data distribution of the training data (70% of the entire dataset) of the BERT model, as observed by classes 1, 2, 3, 4, 5, 9, and 20, which are very unbalanced compared with classes 6, 7, 8, 10, and 11. Therefore, the TABGAN-augmented classes are classes 1, 2, 3, 4, 5, 9, and 20 (as indicated in Table 3). Finally, the augmented data are added to the training data; so, the data to train the BERT+TABGAN network are distributed as illustrated in Figure 7.

Table 5 shows the results obtained when using BERT+TABGAN. As before, the best model reported previously by us in [11] (RF model) continues to deliver better performance (Table 6); however, the new model outperforms the other models reported in that paper, such as SVM and MLP. The advantage of the proposal is that by using BERT and BERT+GAN neural networks it is possible to increase the performance of the activity recognition system compared with other promising machine learning models such as SVM and MLP. In this work, the BERT network is designed with the parameters assigned by default in Python; however, it is possible that modifying these parameters and using BERT models with more layers or testing other types of pre-trained transformers can further improve performance and even exceed the performance of the RF model.

As shown in Table 6, the performance of the BERT+TABGAN model outperforms the performance of the BERT network in all metrics. BERT+TABGAN outranks BERT by 0.36% Accuracy, 7.43% Precision, 5.19% Recall, and 6.25%
$F_{1}$-Score. Therefore, the hypothesis that it is possible to improve the performance of the activity recognition system with the increase in data with TABGAN by means of recurrent neural networks trained with sliding windows of skeleton sequences using UP-Fall is demonstrated.

Figure 8 shows the confusion matrix when using BERT+TABGAN. As with the BERT network, fall activities are often confused with class 11 (lying down).

Finally, Figure 9 compares the confusion matrices of the BERT model with the BERT+TABGAN model. With BERT+TABGAN, the recall of each class increases; so, the system is better at predicting each class and reducing the confusion between them. Except for class 9 (picking up an object), BERT+TABGAN better recognizes all activities.

## 5. Conclusions

This paper presented an activity recognition approach using video data with a classifier model based on transformers using human skeleton poses as inputs and tested using the public dataset UP-Fall to provide a direct comparison with previous work reported in [11]. The presented BERT model outperforms the results obtained by the LSTM, SVM, and MLP models.

The proposed method demonstrated that data augmentation using GAN to generate synthetic data improved model performance since a balanced database adds generalization capability to the classifier. Using an optimization method, it is possible to find the proportion of artificial data that allows obtaining the best performance for the BERT classifier.

Although Table 6 shows that the [11] RF model is still better than our TABGAN, it is important to highlight some advantages of our model over the models delivered in previous works. Thanks to the use of GAN networks combined with BERT, it is possible to use datasets for the recognition of activities, even when the classes are unbalanced or when the amount of data is scarce. On the other hand, the use of BERT networks in vision applications opens up many possibilities, such as combining the use of words and skeletons with the pose information of a person in an image to create new images or videos from only a sentence or to tell stories from images.

Future work is expected to validate with other pre-trained models based on transformers and different GAN architectures to increase detection rates and verify the proposed methodology’s operation with other datasets of multiple people in open spaces.

## Figures and Tables

**Figure 1 sensors-23-01400-f001:**
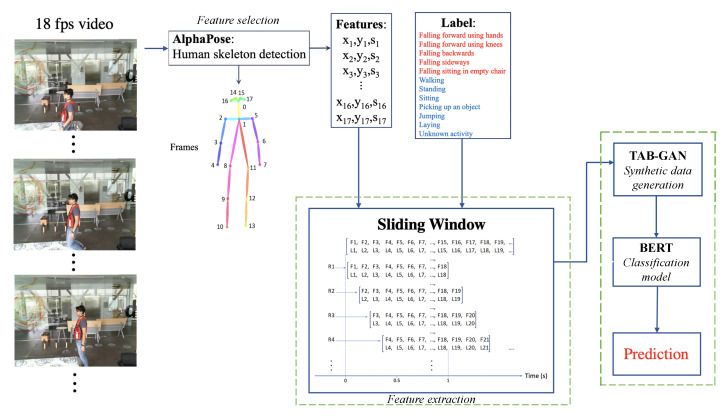
Workflow for activity recognition.

**Figure 2 sensors-23-01400-f002:**
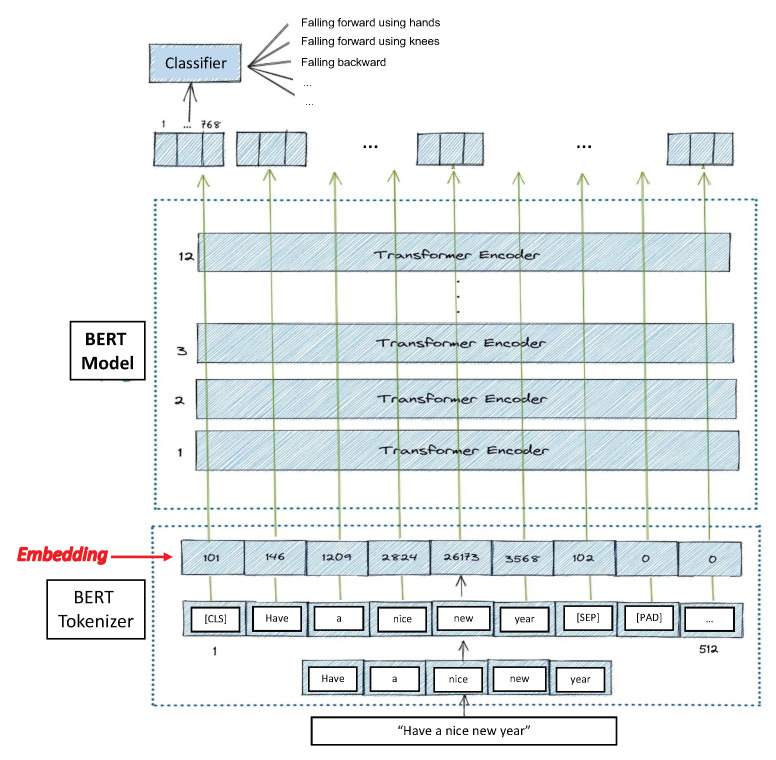
Illustration of the input and output of the BERT model.

**Figure 3 sensors-23-01400-f003:**
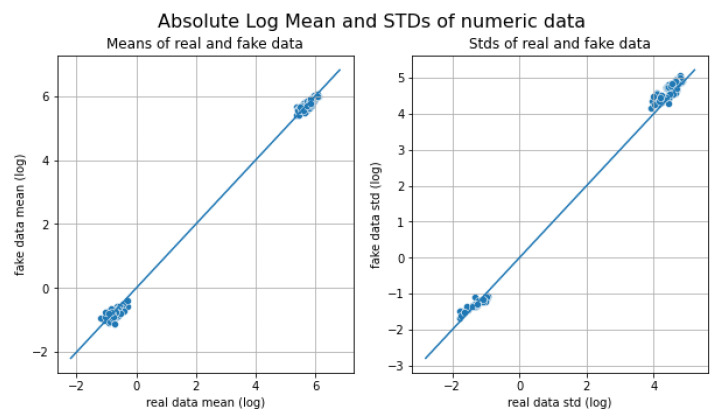
Relationship between the distribution of real data and generated synthetic data: Mean (**left**) and standard deviation (**right**).

**Figure 4 sensors-23-01400-f004:**
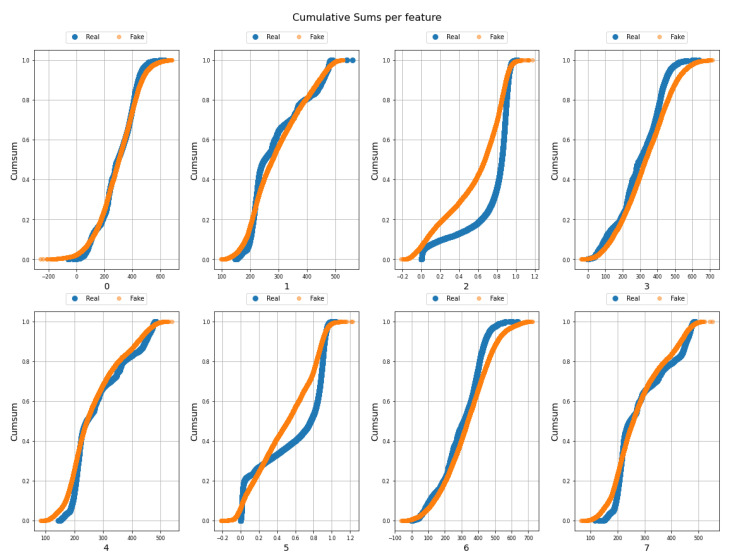
Cumulative sum of real data (blue) and synthetic data (orange) per feature for the first 8 features.

**Figure 5 sensors-23-01400-f005:**
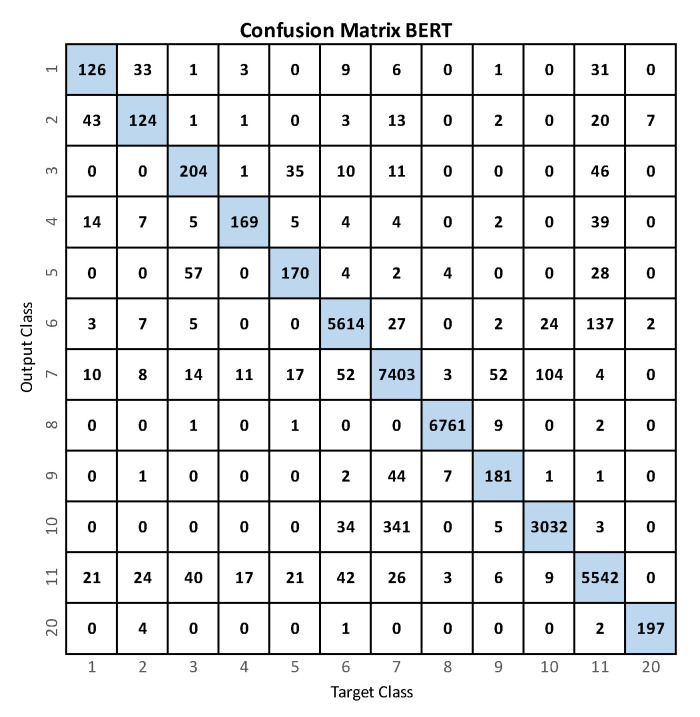
Confusion matrix for the recognition of activities with BERT.

**Figure 6 sensors-23-01400-f006:**
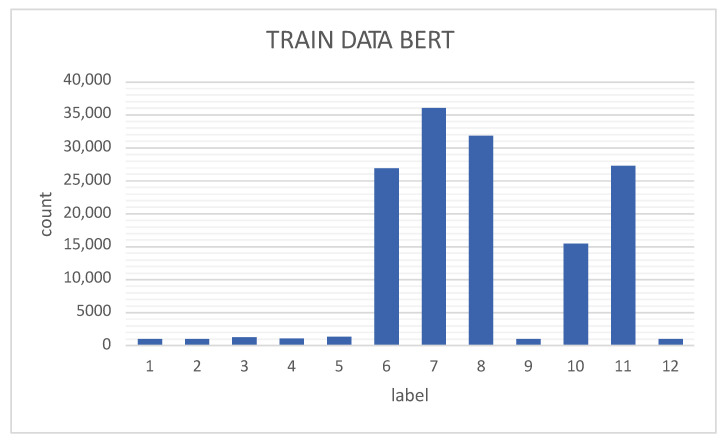
Training data for the BERT model.

**Figure 7 sensors-23-01400-f007:**
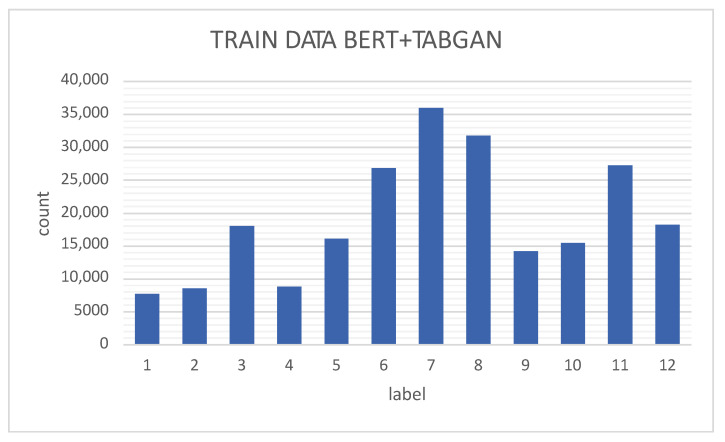
Training data for the BERT+TABGAN model.

**Figure 8 sensors-23-01400-f008:**
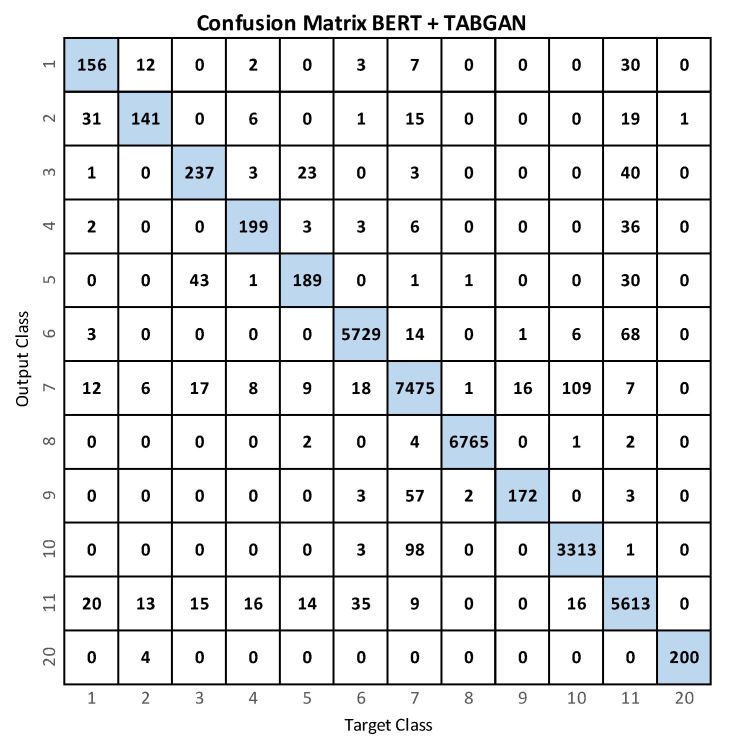
Confusion matrix for activity recognition with BERT+TABGAN.

**Figure 9 sensors-23-01400-f009:**
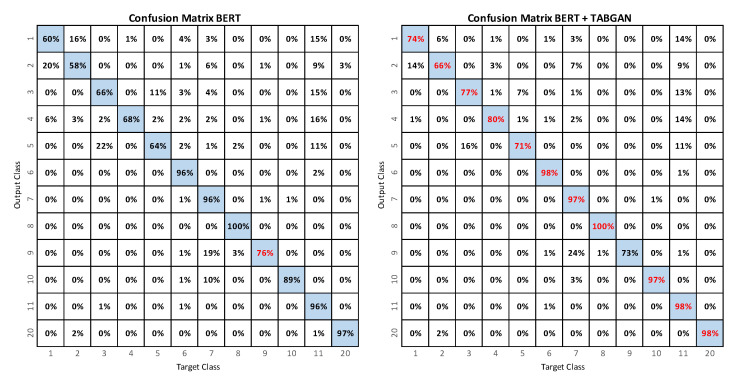
Confusion matrices for activity recognition using BERT (**left**) and BERT+TABGAN (**right**). The best results are shown in red.

**Table 1 sensors-23-01400-t001:** UP-FALL activities.

Label	Activity
1	Falling forward using hands
2	Falling forward using knees
3	Falling backward
4	Falling sideways
5	Falling sitting in an empty chair
6	Walking
7	Standing
8	Sitting
9	Picking up an object
10	Jumping
11	Laying
12	Unknown activity

**Table 2 sensors-23-01400-t002:** Hyperparameters configuration for the BERT model.

Model	Hyperparameters
BERT	Epochs = 20
	Learning Rate = $2\times 10^{-5}$
	Batch size = 16
	Optimizer = AdamW

**Table 3 sensors-23-01400-t003:** Distribution of the dataset by class.

Class	# True	% Dataset	# False	Total	% Dataset
1	1394	0.672	17,267	18,661	6.40
2	1473	0.710	6731	8204	2.81
3	1473	0.710	7533	9006	3.09
4	1858	0.895	16,759	18,617	6.39
5	1560	0.752	7747	9307	3.19
6	1863	0.898	14,784	16,647	5.71
7	38,570	18.588	0	38,570	13.23
8	51,573	24.855	0	51,573	17.69
9	45,439	21.899	0	45,439	15.58
10	1456	0.702	13,244	14,700	5.04
11	22,067	10.635	0	22,067	7.57
12	38,771	18.685	0	38,771	13.30

**Table 4 sensors-23-01400-t004:** Performance (%) obtained by the BERT model of the activity recognition method using UP-Fall.

Performance of the Proposed Method													
**Label**	**1**	**2**	**3**	**4**	**5**	**6**	**7**	**8**	**9**	**10**	**11**	**12**	**Average**
Accuracy	99.44	99.44	99.27	99.64	99.44	98.82	97.59	99.90	99.57	98.33	98.32	99.95	99.14
Precision	58.06	59.62	62.20	83.66	68.27	97.21	93.98	99.75	69.62	95.65	94.65	95.63	81.53
Recall	60.00	57.94	66.45	67.87	64.15	96.44	96.42	99.81	76.37	88.78	96.37	96.57	80.60
Specificity	99.71	99.73	99.60	99.89	99.74	99.36	97.98	99.93	99.74	99.50	98.77	99.97	99.49
$F_{1}$-Score	59.02	58.77	64.25	74.94	66.15	96.83	95.18	99.78	72.84	92.09	95.50	96.10	80.95
Support	210	214	307	249	265	5821	7678	6774	237	3415	5751	204	31,125

**Table 5 sensors-23-01400-t005:** Performance (%) obtained by the BERT+TABGAN model of the activity recognition system using UP-Fall.

Performance of the Proposed Method													
**Label**	**1**	**2**	**3**	**4**	**5**	**6**	**7**	**8**	**9**	**10**	**11**	**12**	**Average**
Accuracy	99.60	99.65	99.53	99.72	99.59	99.49	98.66	99.96	99.74	99.25	98.80	99.98	99.50
Precision	69.33	80.11	75.96	84.68	78.75	98.86	97.22	99.94	91.01	96.17	95.97	99.50	88.96
Recall	74.29	65.89	77.20	79.92	71.32	98.42	97.36	99.87	72.57	97.01	97.60	98.04	85.79
Specificity	99.78	99.89	99.76	99.88	99.83	99.74	99.09	99.98	99.94	99.52	99.97	100	99.71
$F_{1}$-Score	71.72	72.31	76.58	82.23	74.85	98.64	97.29	99.90	80.75	96.59	96.78	98.77	87.20
Support	210	214	307	249	265	5821	7678	6774	237	3415	5751	204	31125

**Table 6 sensors-23-01400-t006:** Comparison between our proposal and the activity recognition models implemented in [11] for the UP-FALL dataset and sliding windows using skeletons.

Model	RF	SVM	MLP	BERT	BERT
					+GAN
	in [11]	in [11]	in [11]	Our	Our
Accuracy	99.91	98.60	99.28	99.14	99.50
Precision	97.73	95.60	82.71	81.53	88.96
Recall	95.60	57.40	78.97	80.60	85.79
Specificity	99.95	99.14	99.58	99.49	99.71
$F_{1}$-Score	96.63	62.87	79.89	80.95	87.20

## Data Availability

The data used for this study is available publicly on https://sites.google.com/up.edu.mx/har-up/ accessed on 20 January 2023.

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
