# Peer review of "BERT for Activity Recognition Using Sequences of Skeleton Features and Data Augmentation with GAN"

_sensors, 2023, doi:10.3390/s23031400_

Round 1
Reviewer 1 Report
1-The main problem of this paper is that this method proposed by itself is less introduced.
2- Many very detailed parameters are missing.
Author Response
Thank you for your review. Please see attached file with our response to all reviewers.

Reviewer 2 Report
The novelty is limited in general. The claimed two novel points are not novel enough. "the use of attention-based deep learning to perform the activity classification task" is very common and not novel. "The generation of these artificial data is done using GANs" is also not novel. And a critical step is how to discrete the features into 30,000 tokens when using BERT, which is not clear in the paper.
Author Response

(The authors gave the same response as above.)

Reviewer 3 Report
The paper proposed an approach for activity recognition based on computer vision. BERT is used for feature extraction and GAN is used for classification. The organization of the paper has to be improved and the contributions of the paper should be better clarified as compared to the previous work. Some of my detailed comments can be found below:
1. In the abstract, it is not clear what is the challenging of activity recognition and what is the state-of-the-art. Attention model is mentioned in the abstract, but it is not explained in the methodology.
2. In Line 57, the authors mentioned that two approaches are proposed in the paper. It is not clear how those approaches are implemented and there is no experimental evaluation of the two approaches.
3. The connections between different sections (for example Section 3.1, Section 3.2, and Section 3.3) are not clear. These sections are the key of the proposed approach and should be highlighted in the workflow in Figure 1.
4. I suggest the authors to provide more details on the datasets used in this paper.
5. Line 97 and Line 104, I suggest to put the link as footnote of the paper.
6. Figure 1 is quite confusing. It is not clear how the model is trained and how the classification is performed.
7. Line 148, should add a reference for “Ramirez et al.”. It is very hard to see the text in Figure 3.
8. In Table 6, what is the advantage of the proposed approach, as RF gives better results when compared to the proposed approach.
9. Table 3 shows 12 classes, starting from 0 to 11, while Table 4 lists 12 classes, starting from 1 to 12. It is not consistent.
10. The organization of the experimental have to be improved. It is very difficult to catch the meaning of the figures.
Author Response

(The authors gave the same response as above.)

Reviewer 4 Report
(1)The abstract should be improved. Your point is your own work that should be further highlighted.
(2)The parameters in expressions are given and explained.
(3) The method in the context of the proposed work should be written in detail
(4) In the 4.1. BERT, the values of parameters could be a complicated problem itself, how the authors give the values of parameters in the BERT, such as "145,247 windows",“31,125 windows”,"8,227","199,270" and so on.
(5) About BERT+TABAGAN, the authors should give the detailed describing.
(6) The literature review is poor in this paper. You must review all significant similar works that have been done. I hope that the authors can add some new references in order to improve the reviews. For example,
https://doi.org/10.1016/j.ymssp.2022.109422ï¼›
https://doi.org/10.3934/mbe.2023090;
https://doi.org/10.3389/fendo.2022.1057089;
https://doi.org/10.1016/j.eswa.2021.114629 and so on.
(7) The authors are requested to correct all spelling mistakes.
Author Response

(The authors gave the same response as above.)

Round 2
Reviewer 2 Report
The authors have provided a revised version which has solved most of my questions. So I am positive about this version now.
Author Response
Many thanks for your prompt review. Please see attached responses.

Reviewer 3 Report
Many typo errors can be found in the paper.
In Table 6, the literature [11] gives better results. Please discuss what is the advantage of the proposed approach.
Author Response

(The authors gave the same response as above.)

Reviewer 4 Report
According to the revised paper, I have appreciated the deep revision of the contents and the present form of this manuscript. There is little content, which need be revised according to the comment of reviewer in order to meet the requirements of publish. A number of concerns listed as follows:
(2) In the introduction section, you should clearly describe the contributions of your works.
(3) How to determine these parameters? The author should give a detailed explanation.
(4) Conclusion: What are the advantages and disadvantages of this study compared to the existing studies in this area?
(5) In order to further highlight the introduction, some suggested references should be added to the paper for improving the reviews part.
Author Response

(The authors gave the same response as above.)
